# Intraoperative Blood Pressure Variability Predicts Postoperative Mortality in Non-Cardiac Surgery—A Prospective Observational Cohort Study

**DOI:** 10.3390/ijerph16224380

**Published:** 2019-11-09

**Authors:** Agnieszka Wiórek, Łukasz J. Krzych

**Affiliations:** Department of Anaesthesiology and Intensive Care, School of Medicine in Katowice, Medical University of Silesia, 14 Medyków Street, 40-752 Katowice, Poland

**Keywords:** blood pressure variability, intraoperative monitoring, postoperative mortality, non-cardiac surgery

## Abstract

Little is known about the clinical importance of blood pressure variability (BPV) during anesthesia in non-cardiac surgery. We sought to investigate the impact of intraoperative BPV on postoperative mortality in non-cardiac surgery subjects, taking into account patient- and procedure-related variables. This prospective observational study covered 835 randomly selected patients who underwent gastrointestinal (*n* = 221), gynecological (*n* = 368) and neurosurgical (*n* = 246) procedures. Patient’s and procedure’s risks were assessed according to the validated tools and guidelines. Blood pressure (systolic, SBP, and diastolic, DBP) was recorded in five-minute intervals during anesthesia. Mean arterial pressure (MAP) was assessed. Individual coefficients of variation (Cv) were calculated. Postoperative 30-day mortality was considered the outcome. Median SBP_Cv was 11.2% (IQR 8.4–14.6), DBP_Cv was 12.7% (IQR 9.8–16.3) and MAP_Cv was 10.96% (IQR 8.26–13.86). Mortality was 2%. High SBP_Cv (i.e., ≥11.9%) was associated with increased mortality by 4.5 times (OR = 4.55; 95% CI 1.48–13.93; *p* = 0.008). High DBP_Cv (i.e., ≥22.4%) was associated with increased mortality by nearly 10 times (OR = 9.73; 95% CI 3.26–28.99; *p* < 0.001). High MAP_Cv (i.e., ≥13.6%) was associated with increased mortality by 3.5 times (OR = 3.44; 95% CI 1.34–8.83; *p* = 0.01). In logistic regression, it was confirmed that the outcome was dependent on both SBPV and DBPV, after adjustment for perioperative variables, with AUCSBP_Cv = 0.884 (95% CI 0.859–0.906; *p* < 0.001) and AUCDBP_Cv = 0.897 (95% CI 0.873–0.918; *p* < 0.001). Therefore, intraoperative BPV may be considered a prognostic factor for the postoperative mortality in non-cardiac surgery, and DBPV seems more accurate in outcome prediction than SBPV.

## 1. Introduction

Blood pressure (BP) is routinely monitored during anesthesia giving insight into organ perfusion in a safe, non-invasive way [1,2]. It also allows to implement goal-directed hemodynamic therapy, including proper fluid regimen and catecholamines [2,3]. 

The effects of perioperative hypo- and hypertension on postoperative mortality have been extensively studied, both in cardiac and non-cardiac surgeries [4,5,6,7]. However, there is still a paucity of data regarding clinical importance of BP variations in non-cardiac surgery; though the number of studies focusing on ambulatory BP variability among non-surgical patients has risen in the last decade [2]. Firstly, perioperative variations may result from multiple patient- and environment-related factors. Secondly, they may complicate perioperative homeostasis in a complex way. Improper hypotensive treatment in the preoperative period increases the rate of cardiovascular adverse events [8]. Unsuitable monitoring (i.e., when the patient’s treatment is not based on hemodynamic monitoring) and unsuitable treatment of BP may counterfeit inadequate levels of anesthesia or analgesia delivered during surgery due to individual susceptibility or tolerance for the administered drugs. Additionally, massive bleeding or any other type of shock has a significant influence on BP changes.

Because further research has been recommended to draw reasonable conclusions on this issue [9,10,11], we aimed to assess the impact of intraoperative BP variability on postoperative mortality in non-cardiac subjects, taking into account the confounding effect of patient- and procedure-related risk. We hypothesized that the higher the BP variability, the worse the outcome regardless of the origin of the variations.

## 2. Materials and Methods 

In this prospective observational cohort study, we covered 835 patients who underwent surgery between 1 January 2017 and 31 December 2017 in a tertiary university hospital, including gastrointestinal (*n* = 221), gynecological (*n* = 368) and neurosurgical (*n* = 246) procedures. To select a representative sample from all subjects being anaesthetized (i.e., underwent general and/or regional anesthesia) in 2017, we used a multistage approach, which was a combination of cluster and systematic sampling (i.e., all consecutive subjects anesthetized every third week of each month were included). No a priori power or sample calculation was performed. None of the subjects were missed in the screening period. The selection process of the participants is depicted in Figure 1.

Demographic and medical data were recorded at the time of the surgery, including sex, age, body mass index (BMI), the patient’s risk category according to the American Society of Anesthesiology (ASA) physical status (PS) classification [12], type and the duration of anesthesia, the position of patients in which the procedure was performed, whether the surgery was performed as elective, urgent or emergency, and the risk of the procedure according to the ESC/ESA guidelines [13]. ASA-PS VI patients (i.e., organ donors) were excluded from the analysis. No other exclusions were applied. 

Under Section 21 and 22 of the Act of 5 December 1996 on the Medical Profession [14], due to the non-interventional design of the study, no approval of the Ethics Committee was required. However, all patient data were obtained in accordance with the national law regulations of personal data management, after the written consent was given by the patients on hospital admission, excluding unconscious patients who required emergency procedures.

All patients had multiple blood pressure measurements performed at the time of the surgery by the Primus® integrated anesthesia workstation (Draeger®, Germany) equipped with an automated oscillometric non-invasive BP monitoring (Marey technique) [15]. Systolic blood pressure (SBP) and diastolic blood pressure (DBP) values were recorded in five-minute intervals during anesthesia (from the first pre-induction measurement until the last measurement during recovery from anesthesia in the operating theatre). Mean arterial pressure (MAP) values were subsequently calculated. Coefficients of variation (Cv, (%)) were calculated to express BP variability (BPV). Cvs were calculated based on the equation Cv = S/X*100%, where “S” stands for standard deviation and “X” stands for the arithmetic means of all measurements taken. The ROC analysis was performed to statistically assess the optimal cut-off points in outcome prediction. Then, the cut-off point values of BPV_Cv for SBP, DBP and MAP were applied to distinguish categories of low and high BPV.

Postoperative 30-day mortality was considered the outcome.

STROBE Statement was applied for appropriate data reporting.

Statistical analysis was performed using MedCalc v.18 software (MedCalc Software, Ostend, Belgium). Quantitative variables were depicted using medians and interquartile ranges (IQR, i.e., 25pc–75pc). The Shapiro-Wilk test was used to verify their distributions. Qualitative variables were described with frequencies and percentages. Between-group differences for continuous variables were assessed using the Kruskal-Wallis test, while for categorical variables the Chi-squared test was applied. Odds ratios (OR) with their 95% confidence intervals (CI) were also calculated to express the associations between high BPV and mortality. The correlation was assessed using Spearman’s rank correlation coefficient. Logistic regression was performed to verify observations from bivariate models, taking into account the confounding effect of patient- and procedure-related risk (see Table 1). Variables with a *p*-value < 0.1 in bivariate comparisons were consecutively subjected to a multiple regression model. BPV was considered the exposure variable, patient- and procedure-related data were confounding variables, and the outcome variable was mortality. Logistic ORs (95%CI) were subsequently estimated. ROC curves were drawn and areas under the ROC curves (AUC) were calculated to determine the predictive value of BPV and the outcome. AUC’s were also calculated to assess the diagnostic accuracy of the final logistic regression equations. A *p*-value < 0.05 was considered significant.

## 3. Results

The study group consisted of 835 patients (231 men, 28%). The median age was 48 years (IQR 34–62). The median time of anesthesia was 100 minutes (IQR 60–160). Detailed characteristics of the study group and procedure-related parameters are depicted in Table 1. The median of SBP_Cv was 11.24% (IQR 8.40–14.59), it was 12.68% (IQR 9.79–16.33) for DBP_Cv, and 10.96% (IQR 8.26–13.86) for MAP_Cv. 

In bivariate investigations, SBP_Cv statistically differed significantly in terms of sex, ASA-PS class, type of anesthesia, type of procedure, surgical position and the procedure’s risk category. DBP_Cv and MAP_Cv differed in terms of ASA-PS class, type of anesthesia, type of procedure, surgical position and the procedure’s risk category (Table 2). In addition, BP_Cv statistically significantly but weakly correlated with age, BMI and duration of anesthesia (Table 3).

Postoperative 30-day mortality was 2% (*n* = 18). SBP_Cv, DBP_Cv and MAP_Cv were higher among the deceased patients (Figure 2, Figure 3 and Figure 4). 

SBP_Cv predicted mortality in a statistically significant way, with AUC = 0.705 (95% CI 0.672–0.735, *p* < 0.001) at the cut-off point of 11.92% (sensitivity 77.8%, specificity 56.6%) (Figure 5). DBP_Cv also predicted the outcome, with AUC = 0.638 (95% CI 0.605–0.671, *p* = 0.04) at the cut-off point of 22.4% (sensitivity 27.8%, specificity 96.3%) (Figure 6). Finally, MAP_Cv predicted mortality in a statistically significant way, with AUC = 0.655 (95% CI 0.622–0.688, *p* = 0.024) at the cut-off point of 13.6% (sensitivity 53%, specificity 73.6%) (Figure 7).

High SBP_Cv (i.e., ≥11.92%) was found in 369 subjects and high DBP_Cv (i.e., ≥22.4%) in 36 patients. Both high BPVs were diagnosed in 34 persons. High MAP_Cv (i.e., ≥13.6%) was observed in 227 patients. High SBP_Cv was associated with increased mortality by 4.5 times (OR = 4.55; 95% CI 1.48–13.93; *p* = 0.008). High DBP_Cv was associated with increased mortality by nearly 10 times (OR = 9.73; 95%CI 3.26–28.99; *p* < 0.0001). Both high BPVs were associated with increased mortality by 10 times (OR = 10.42; 95%CI 3.48–31.19; *p* < 0.0001). High MAP_Cv was associated with increased mortality by 3.5 times (OR = 3.44; 95% 1.34–8.83; *p* = 0.01).

In logistic regression, it was confirmed that after adjustment for patient- and procedure-related variables, postoperative mortality was dependent on systolic and diastolic BPV (both logistic ORs = 1.10). However, the association for SBP was of borderline significance (Table 4). The association for MAP was not significant (*p* = 0.06). The overall diagnostic accuracy was very good, with AUC = 0.884 for SPV and AUC = 0.897 for DBP.

## 4. Discussion

Our study aimed to assess the impact of intraoperative BPV on postoperative mortality in the non-cardiac surgery setting, with the consideration of the confounding effect of patient- and procedure-related risk. The risk was calculated on the basis of widely used, clinically established, and standardized methods. Mortality was only 2%, however, we demonstrated that BP variations were clearly among the factors increasing the risk of post-operative death, with a high diagnostic accuracy of the statistical model. This is in line with current suggestions to incorporate BPV into existing risk scores [16]. Interestingly, DBPV was more accurate than SBPV or mean BPV in outcome prediction. 

There have been several studies investigating the impact of intraoperative trends of decreased or increased BP values on the risk of death. In non-cardiac operations, Spaite et al. analyzed outcomes of patients who underwent surgery due to severe traumatic brain injury and revealed that an episode of intraoperative hypotension considerably increased the risk of death [4]. They also found a linear relationship between systolic BP in the early phase of the injury and subsequent mortality [4]. In the cardiac-surgery setting, Maheshwari and colleagues showed that low intraoperative BP was associated with increased all-cause in-hospital mortality [6]. Balzer et al. confirmed that perioperative hypertension was associated with compromised outcomes, including higher mortality rate [7]. 

Comparison of our results with literature data is limited because our study is relatively novel in the field. Our observations vary slightly from those of Venkatesan et al. who revealed that only DBP variability was associated with an increased risk of mortality [17]. Nevertheless, in our analysis, high DBPV predicted mortality better than SBPV. Additionally, Monk et al. [5], while investigating the relationship between mean arterial pressure (MAP) percent change from the baseline and 30-day mortality, found that if MAP decreased to more than 50% for ≥5 min during the surgery, the mortality increased by nearly three times. Interesting and contradictory data come from the study of Mascha et al. [18] who investigated the role of lability of MAP in a retrospective cohort of over 100,000 patients undergoing non-cardiac procedures lasting over 60 minutes. They revealed that low BPV was associated with higher 30-day mortality and concluded that anesthesiologists should rather pay more attention to overall trends in the MAP for a case than in the minute-to-minute variation. In addition, Levin et al. found that for each episode of lability of mean BP >10% decreased the 30-day mortality by 4%–5% (*p* < 0.01) but only in patients taking no antihypertensive medication (with or without a history of hypertension) [11]. The authors concluded that labile hemodynamic responses might reflect an intact autonomic nervous system that is adapting appropriately to the stress of surgery. These discrepancies are difficult to explain, but based on observations of Vernooij et al. in non-cardiac surgery [19], as well as those of Sun et al. in cardiac surgery operations [20], we suppose they may arise from a method of BPV assessment and its definition, as well as from differences between the groups under investigation. Vernooij with colleagues applied eight definitions of intraoperative hypotension, as well as twelve different methods representing presence, depth, duration, and area under the threshold of hypotension to verify the association between BP drop and the occurrence of postoperative myocardial injury or acute kidney injury [19]. In their investigations, methods with the highest odds were an absolute maximum decrease in BP and mean episode area under the threshold of 50 mmHg of MAP [19]. Sun et al. investigated the association between stroke and hypotension using three BP categories before, during, and after cardiopulmonary bypass [20]. They found that perioperative stroke was more frequent in patients with MAP lower than 64 mmHg for at least 10 minutes [20]. Thus, further research needs to focus not only on absolute BP values during surgery or BP fluctuations, but also on the duration of organ hypoperfusion. 

Although BP is one of the basic vital signs monitored perioperatively, finding the optimal BP target during anesthesia (i.e., when sympathetic system activity is reduced) remains troublesome [21]. Numerous thresholds for systolic, diastolic and mean BP have been suggested to maintain suitable organ perfusion, being usually within 20% of the preoperative values [1]. Most studies have investigated clinical effects of BP variations on regional perfusion, including cardiac, renal and neurological side effects associated with BP instability [1]. Those examinations primarily focused on the detrimental role of hypotension on the myocardial injury, acute kidney injury and brain insult causing disturbances in the consciousness [18,19,20,22,23]. Thus far, our project is the first research attempted to establish a ‘safe’ BPV level during the surgery [24].

### Study Limitations

One should bear in mind possible limitations regarding the presented study. Firstly, it is observational in design, and interventional investigation should be performed to confirm our findings. In addition, this a single-center observation. Secondly, there was a relatively low death rate in our cohort so the detailed analysis of mortality was impossible to be performed. However, as no power analysis was performed to assess the sample size, one ought to realize the potential selection bias. Thirdly, the first BP measurement was performed in the operating room before induction of anesthesia, which may impact the magnitude of BPV. Limitations come also from the non-invasive technique of BP monitoring, in which the correct measurement method is crucial to obtain reliable results [14,25]. However, this method is convergent with current recommendations for standards of monitoring during anesthesia [26]. Fourthly, intraoperative fluid therapy should be taken into account in data interpretation. In our patients, we used MAP-guided goal-directed therapy to adjust the dose of fluids to maintain hemodynamic equilibrium. Thus, the fluid regimen was tailored strictly to the patients’ needs and requirements. It is therefore obvious that precise monitoring of BP lability could facilitate the maintenance of well-balanced perioperative fluid and vasopressor therapy, decreasing the risk of adverse effects of higher-than-needed vasopressor dosage [27]. We believe that inappropriate tissue oxygenation, regardless of its origin, gives similar effects in terms of increasing mortality in the perioperative period. DBP may be however more accurate than SBP to detect abnormal tissue perfusion. Finally, our analysis did not cover the incidence of preoperative anemia, intraoperative blood loss and the strategy of perioperative transfusions. Although patients with anemia had higher BPV (data not shown), this association was insignificant after adjustment for ASA-PS classes. However, basing on the recent observations that anemic patients have an increased risk of postoperative complications and death [28], the association between anemia and BPV deserves further investigations. Interestingly, Fowler with co-workers revealed that 30.1% of adults undergoing elective inpatient surgery were anaemic. Patients with moderate (i.e., hemoglobin 8–10.9 g·dL^-1^) and severe anemia (i.e., hemoglobin <8 g·dL^-1^) were at an increased risk of death by three times and four times, respectively [28], after adjustment for basic demographic factors, ASA-PS classes and procedure-related variables. These findings shed light on the importance of preoperative correction of anemia, especially in subjects at high risk of intraoperative blood loss. 

## 5. Conclusions

We may conclude that even after adjustment for the preoperative risk, intraoperative BPV may be considered a prognostic factor for the postoperative mortality in non-cardiac surgery, and DBPV seems more accurate in mortality prediction than SBPV. Generalizability of those results is limited, so further research is needed to determine the beneficial role of early personalized hemodynamic therapy to maintain BP stability during anesthesia. 

## Figures and Tables

**Figure 1 ijerph-16-04380-f001:**
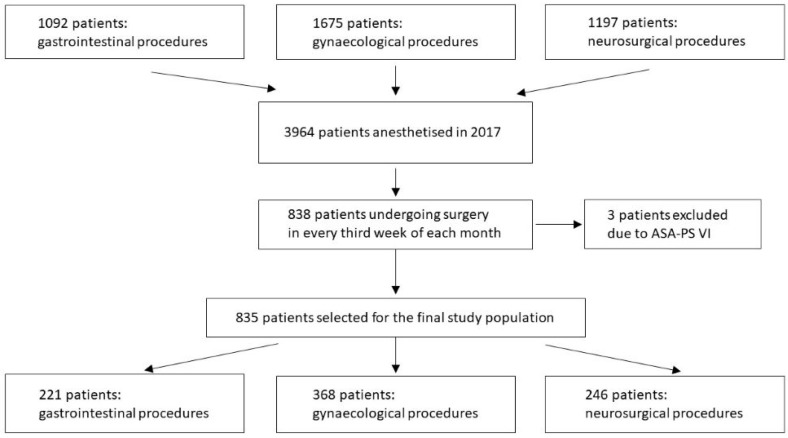
Flowchart of the study population.

**Figure 2 ijerph-16-04380-f002:**
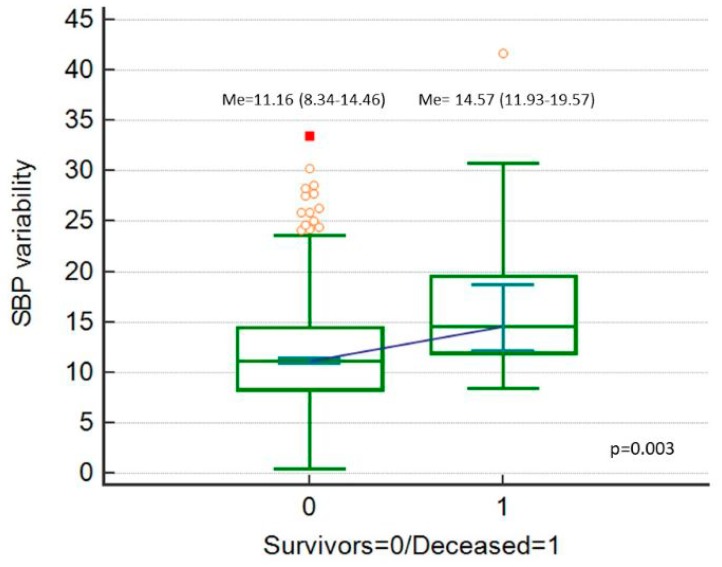
Systolic blood pressure (SBP) variability (expressed as SBP_Cv in %), comparison between deceased patients and survivors.

**Figure 3 ijerph-16-04380-f003:**
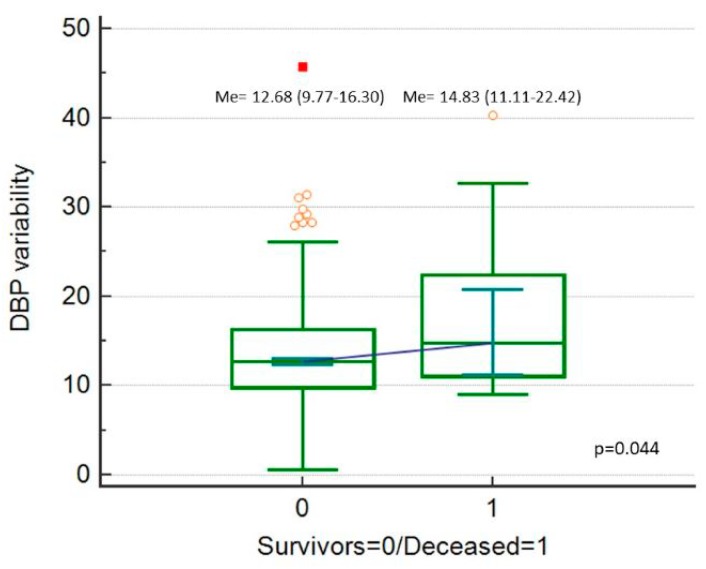
Diastolic blood pressure (DBP) variability (expressed as DBP_Cv in %), comparison between deceased patients and survivors.

**Figure 4 ijerph-16-04380-f004:**
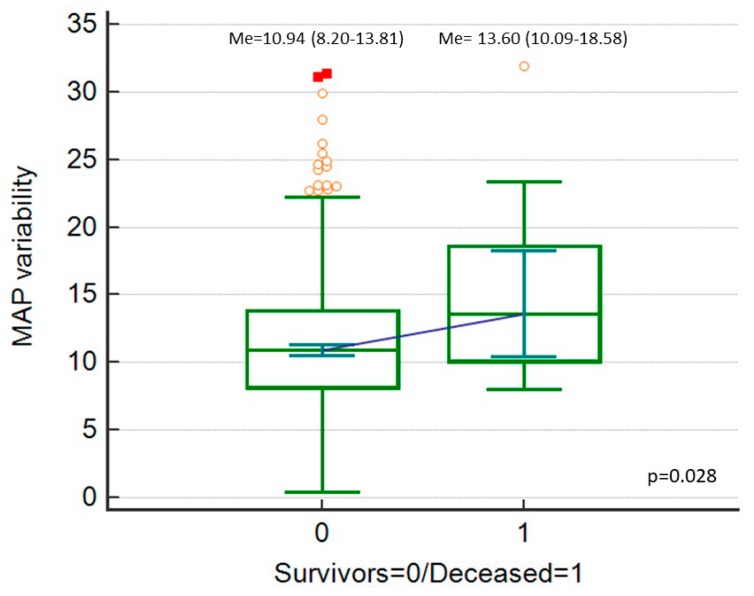
Mean blood pressure (MAP) variability (expressed as MAP_Cv in %), comparison between deceased patients and survivors.

**Figure 5 ijerph-16-04380-f005:**
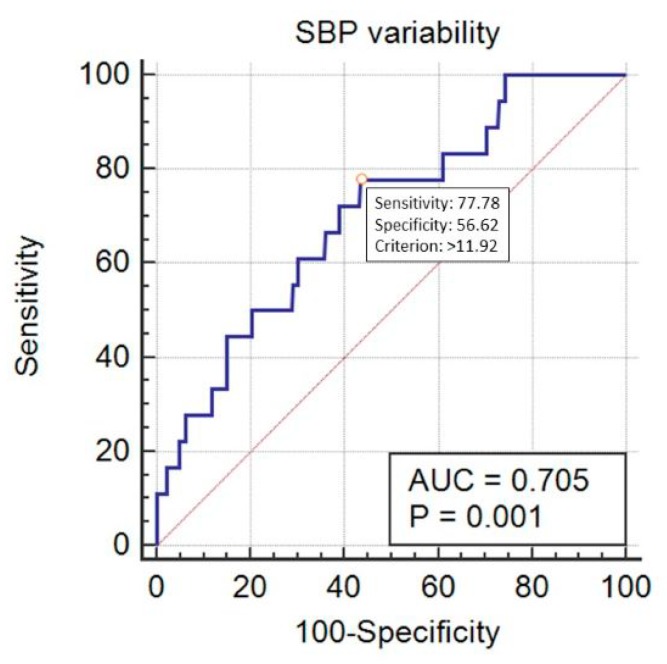
Prediction of mortality by the systolic blood pressure (SBP) variability (expressed as SBP_Cv in %).

**Figure 6 ijerph-16-04380-f006:**
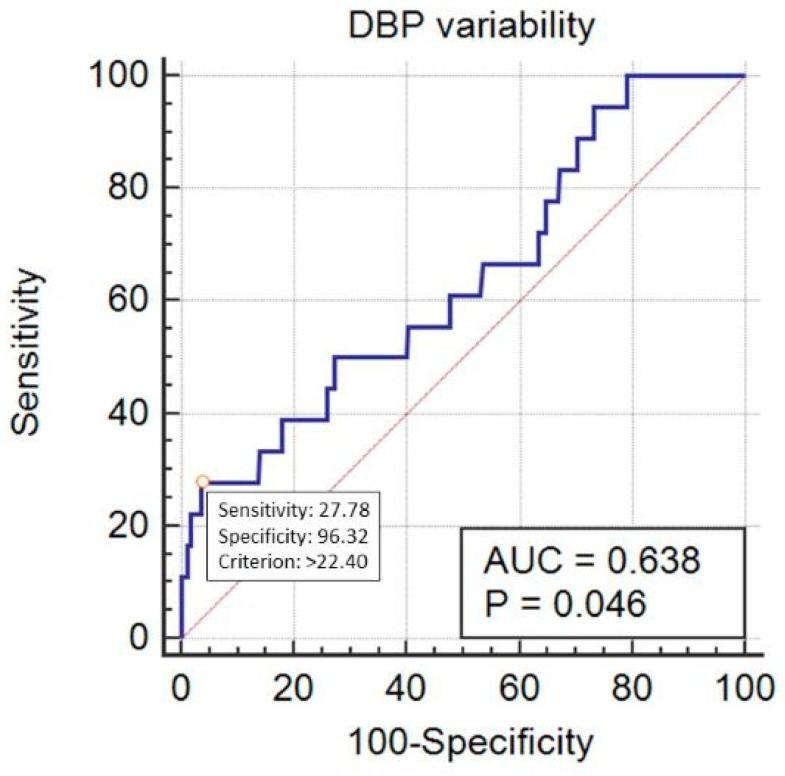
Prediction of mortality by the diastolic blood pressure (DBP) variability (expressed as DBP_Cv in %).

**Figure 7 ijerph-16-04380-f007:**
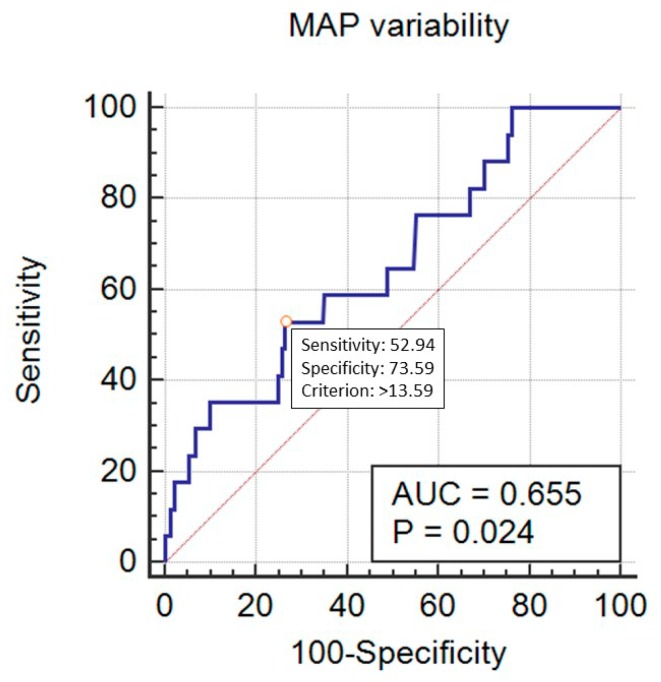
Prediction of mortality by the mean blood pressure (MAP) variability (expressed as MAP_Cv in %).

**Table 1 ijerph-16-04380-t001:** Study group characteristics and procedure-related variables ^1^.

Variable	Category	Value
Sex	Males	231 (27.7%)
	Females	604 (72.3%)
Age	(years)	48 (34–62)
Body Mass Index	(kg/m^2^)	26.0 (22.9–29.6)
American Society of Anesthesiology Physical Status (ASA-PS) class	Median class (IQR)	2 (2–3)
	Class I	204 (24.4%)
	Class II	356 (42.6%)
	Class III	203 (24.3%)
	Class IV	65 (7.8%)
	Class V	7 (0.8%)
Urgency of surgery	Emergency surgery	137 (16.4%)
	Non-emergency surgery (elective + urgent)	698 (83.6%)
Length of anaesthesia	(minutes)	100 (60–160)
Type of anaesthesia	Volatile	556 (66.6%)
	TIVA	92 (11.0%)
	Regional or combined (regional + general)	187 (22.4%)
Surgical position	Supine	350 (41.9%)
	Gynecological	352 (42.1%)
	Prone	82 (9.8%)
	Supine + lateral	24 (2.9%)
	Sitting	13 (1.6%)
	Lateral	9 (1.1%)
Risk of the procedure according to ESC/ESA	Low	243 (29.1%)
	Intermediate	509 (61.0%)
	High	83 (9.9%)

^1^ Values are medians, interquartile ranges (Q1–Q3) for quantitative variables, and frequencies and percentages for qualitative variables. ASA-PS—American Society of Anesthesiology Physical Status, ESA—European Society of Anaesthesiology, ESC—European Society of Cardiology, IQR—interquartile range, TIVA—total intravenous anesthesia.

**Table 2 ijerph-16-04380-t002:** Blood pressure variability in categories defined by the patient- and procedure-related variables ^2^.

Variable	Category	SBP_Cv (%)	DBP_Cv (%)	MAP_Cv (%)
Sex	Males	11.45 (8.88–14.70)	12.48 (10.15–16.07)	10.58 (8.68–14.04)
	Females	11.10 (8.08–14.47)	12.76 (9.59–16.43)	11.10 (7.98–13.78)
	‘*p*’	0.042	0.803	0.740
ASA-PS class	Class I	9.34 (6.68–12.07)	11.79 (8.27–14.77)	9.67 (6.91–12.55)
	Class II	10.99 (8.08–13.94)	12.40 (9.61–16.07)	10.58 (8.32–13.68)
	Class III	13.43 (10.51–15.89)	13.66 (11.06–17.02)	12.12 (9.76–14.78)
	Class IV	13.72 (9.97–17.27)	13.70 (11.05–18.39)	12.55 (9.44–16.10)
	Class V	10.13 (9.24–29.75)	17.81 (12.28–31.33)	13.31 (7.71–30.36)
	‘*p*’	<0.001	<0.001	<0.001
Urgency of surgery	Emergency	11.00 (8.49–14.73)	12.75 (10.43–17.15)	11.05 (8.30–13.85)
	Other	11.28 (8.39–14.59)	12.68 (9.77–16.19)	10.94 (8.25–13.85)
	‘*p*’	0.8	0.4	0.499
Type of anaesthesia	TIVA	8.04 (4.82–11.13)	8.98 (5.76–13.01)	7.45 (4.41–11.15)
	Other	11.55 (8.81–14.89)	13.01 (10.30–16.51)	11.35 (8.81–14.13)
	‘*p*’	<0.001	<0.001	<0.001
Type of surgery	Gastrointestinal	12.32 (9.60–14.64)	12.86 (10.47–16.23)	11.15 (9.00–13.77)
	Gynecological	9.86 (6.95–13.29)	11.98 (8.46–15.84)	10.39 (6.94–13.29)
	Neurosurgery	12.32 (9.27–16.02)	13.35 (10.62–17.40)	11.53 (8.94–15.05)
	‘*p*’	<0.001	<0.001	<0.001
Surgical position	Sitting/gynecological	9.78 (6.97–12.97)	11.89 (8.55–15.87)	10.37 (6.97–13.37)
	Horizontal	12.38 (9.47–15.46)	13.06 (10.48–16.65)	11.42 (9.00–14.47)
	‘*p*’	<0.001	<0.001	<0.001
Risk of the procedure	Low	8.84 (6.13–12.15)	11.24 (7.11–15.20)	9.65 (5.99–13.10)
	Intermediate	11.69 (9.09–14.93)	13.22 (10.60–16.51)	11.41 (8.88–14.10)
	High	12.78 (10.93–15.46)	13.36 (10.43–15.93)	11.57 (9.31–13.74)
	‘*p*’	<0.001	<0.001	<0.001

^2^ Values are medians and IQR. ’*p*’ values refer to differences in strata defined by consecutive variables (e.g. sex, ASA-PS class, etc.). ASA-PS—American Society of Anesthesiology Physical Status, DBP_CV—coefficient of variation for diastolic blood pressure (diastolic blood pressure variability), MAP_Cv—coefficient of variation for mean blood pressure (mean blood pressure variability), SBP_Cv—coefficient of variation for systolic blood pressure (systolic blood pressure variability).

**Table 3 ijerph-16-04380-t003:** Correlations between BPV_Cv and quantitative data ^3^.

Variable	SBP_Cv (%)	DBP_Cv (%)	MAP_Cv (%)
Age (years)	R = 0.386 *p* < 0.0001	R = 0.125 *p* = 0.0003	R = 0.217 *P* < 0.0001
BMI (kg/m^2^)	R = 0.130 *p* = 0.0001	R = 0.103 *p* = 0.004	R = 0.141 *P* < 0.001
Duration of anaesthesia (min)	R = 0.228 *p* < 0.0001	R = 0.156 *p* < 0.0001	R = 0.106 *P* = 0.002

^3^ Values are Spearman’s rank of coefficients of correlation. BMI—body mass index, DBP_Cv—coefficient of variation for diastolic blood pressure (diastolic blood pressure variability), MAP_Cv—coefficient of variation for mean blood pressure (mean blood pressure variability), SBP_Cv—coefficient of variation for systolic blood pressure (systolic blood pressure variability).

**Table 4 ijerph-16-04380-t004:** Impact of blood pressure variability on postoperative mortality in logistic regression.

Variable	Mortality Prediction by SBP Variability	Mortality Prediction by DBP Variability	Mortality Prediction by MAP Variability
SBP_Cv (per 1%)	OR = 1.10; 95% CI 1.00–1.21; *p* = 0.05	-	-
DBP_Cv (per 1%)	-	OR = 1.10; 95% CI 1.01–1.21; *p* = 0.03	-
MAP_Cv (per 1%)	-	-	OR = 1.10; 95% CI 0.99–1.23, *p* = 0.06
Sex (Female = 0/Male = 1)	OR = 1.05; 95% CI 0.34–3.28; *p* = 0.9	OR = 1.04; 95% CI 0.33–3.28, *p* = 0.9	OR = 1.00; 95% CI 0.31–3.21, *p* = 0.9
Age (per 1 year)	OR = 1.01; 95% CI 0.97–1.05; *p* = 0.7	OR =1.01; 95% CI 0.98–1.05; *p* = 0.4	OR=1.01; 95% CI 0.97–1.05, *p* = 0.5
BMI (per 1 kg/m^2^)	OR = 0.90; 95% CI 0.81–1.00; *p* = 0.05	OR = 0.894; 95% CI 0.80–1.00; *p* = 0.04	OR = 0.897; 95% CI 0.8–1.0, *p* = 0.05
Patient’s risk (ASA-PS class I–V)	OR = 2.21; 95% CI 1.02–4.79; *p* = 0.04	OR = 2.10; 95% CI 0.97–4.53; *p* = 0.06	OR = 2.12; 95% CI 0.96–4.69, *p* = 0.06
Urgency of surgery (Emergency=1/Non-emergency = 0)	OR = 2.61; 95% CI 0.64–10.70; *p* = 0.18	OR = 2.80; 95% CI 0.69–11.32; *p* = 0.1	OR = 2.66; 95% CI 0.62–11.3, *p* = 0.18
Duration of anesthesia (per 1 min)	OR = 1.00; 95% CI 0.99–1.01; *p* = 0.3	OR = 1.00; 95% CI 0.99–1.01; *p* = 0.4	OR = 1.00; 95% CI 0.99–1.01, *p* = 0.2
Type of anesthesia (TIVA = 1/Other = 0)	OR = 1.09; 95% CI 0.12–10.01; *p* = 0.9	OR = 1.10; 95% CI 0.12–10.30; *p* = 0.9	OR = 1.07; 95% CI 0.11–1.05, *p* = 0.9
Surgical position (Sitting or gynecological = 0/Horizontal = 1)	OR = 1.55; 95% CI 0.28–8.56; *p* = 0.6	OR = 1.56; 95% CI 0.28–8.64; *p* = 0.6	OR = 1.59; 95% CI 0.28–8.95, *p* = 0.6
Procedure’s risk (Low = 0/Intermediate = 1/High = 2	OR = 1.02; 95% CI 0.21–5.04; *p* = 0.9	OR = 1.15; 95% CI 0.23–5.74; *p* = 0.8	OR = 0.93; 95% CI 0.18–4.79, *p* = 0.9
AUC for the final logistic model	0.884; 95% CI 0.859–0.906; *p* < 0.0001	0.897; 95% CI 0.873–0.918; *p* < 0.0001	0.892; 95% CI 0.867–0.913 0.0003

ASA-PS—American Society of Anesthesiology Physical Class, AUC-area under the ROC curve, BMI—body mass index, DBP_Cv—coefficient of variation for diastolic blood pressure (diastolic blood pressure variability), MAP_Cv—coefficient of variation for mean blood pressure (mean blood pressure variability), SBP_Cv—coefficient of variation for systolic blood pressure (systolic blood pressure variability), TIVA—total intravenous anesthesia.

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
