# Peer review of "Intraoperative Blood Pressure Variability Predicts Postoperative Mortality in Non-Cardiac Surgery—A Prospective Observational Cohort Study"

_ijerph, 2019, doi:10.3390/ijerph16224380_

Round 1

Reviewer 1 Report

Thank you for the opportunity to review a very interesting article: “ Intraoperative blood pressure variability predicts 2 postoperative mortality in non-cardiac surgery – a 3 prospective observational cohort study.”

The problem of blood pressure changes during surgery is very timely problem, and however it is extensively, we still do not have clear answers to the problem.

Please see below comments to the authors:

Abstract:

line 17 - please be more precise with specifying the outcome - (30 day postoperative mortality? line 19, 20 - you should also specify what does it mean ‚high’ SBP and DBP_CV - what does high mean? Show the thresholds/cut off points. line 22 - skip word ‚even’ 25, 26 - these conclusions are not self explanatory - why do you state that DBP_CV was more sensitive to organ hyperperfusion? there is nothing about it in the abstract. Please keep with the defined outcomes (i.e. mortality).

the same in the conclusions.

Introduction:

explain the shortcuts under the tables/graphs:

examples - explain under the figures the shortcuts ASA-PS VI, TIVA

Materials and Methods

Figure 1 - flow chart - how many patients in the recruiting weeks were missed and not included in the study? Did you use screening logs? line 72- elective and emergency - what about urgent surgeries? Where there counted as emergent? I undertand that the first measurement of blood pressure was performed pre-induction of anesthesia already in the operating room? If it is so, I would add this as limitation, as this blood pressure measurement very often does not correlate with the baseline blood presure of the patient. - line 86- please be more precise how exactly you decided which values where high or low. In your manuscript you state that:

‚The ROC curve cut-off point values 85 of BPV_Cv for SBP and DBP were applied to distinguish categories of low and high BPV’.

Please describe methods of chosing the cut-of point.

line 87 - main outcome - Postoperative mortality (i.e. a number of deaths during index hospitalisation or within 30 days) was considered the outcome. - I guess 30-day mortality is enough? line 100 - specify the confounding variables

Discussion

I do not understand fully the limitation section sentence:

 ‘Secondly, while the study group seems sufficient in terms of its quality and quantity to draw reasonable conclusions about BPV thresholds, a relatively low death rate impedes the mortality-directed analysis. “

In my understanding all analyses are based on the mortality (main outcome). Please be more precise or rephrase.

Author Response

Reviewer #1.

We appreciate your detailed assessment of our paper and laudatory comments regarding the novelty of our study.

We described the outcome more homogenously and explained definitions of BPV in more details, as requested. We modified the conclusion regarding DBPV. We explained the shortcuts under the tables and figures. None of the subjects were missed so there was no need to modify the flowchart. There was no need to use screening logs. We included all consecutive subjects in subsequent weeks, with no further exclusions. A list of those subjects was a frame. We clarified it in ‘Methods’. Urgent surgeries were included into non-E procedures. We clarified it. Yes, the first measurement was performed before induction of anaesthesia. We included it as study limitation. We clarified the cut-offs of low/ high values. We described methods of choosing the threshold in ‘Methods’ more precisely. We specified which variables were the confounders. We rephrased/ rewrote the discussion.

Kind regards, the Authors

Reviewer 2 Report

October 2nd, 2019                                           

Editor-in-Chief

IJERPH

MDPI                                                

Dear Editor,

Thank you for giving me the opportunity to review the manuscript titled: “Intraoperative blood pressure variability predicts postoperative mortality in non-cardiac surgery – a prospective observational cohort study” submitted by A. Wiórek and Ł. Krzych.

The problem investigated by the authors is important, therefore their effort should be appreciated. The manuscript is written well, yet some points need improvement or clarification. The manuscript needs minor spelling and style improvement.

The aim of this study was to investigate the impact of intraoperative blood pressure variability on outcome, namely postoperative mortality, in patients undergoing non-cardiac procedures. This study included a large group of patients (n=835) undergoing gastrointestinal, gynecological and neurosurgical procedures with a fair representation of patients in each group.

Abstract: It is well written and adapted to the STROBE guidelines. The parts included in the abstract are adequate. The following changes are necessary:

The second sentence in the abstract should be rephrased “…mortality in non-cardiac subjects…” is incorrect. There are many “cardiac” patients undergoing non-cardiac procedures, so here the procedure is non-cardiac, but not the patient. The abstract should include a definition of high SBP_Cv or DBP_Cv based on the cut-off of the ROC curve included in the main text.

Introduction:  It is well-written, minor changes are necessary:

Reference 5 and reference 7 refer to cardiac surgery and this article is aimed at non-cardiac operations. I believe the authors should underline the lack of non-cardiac studies and comment on cardiac surgery in a separate paragraph, with the important influence of the cardio-pulmonary bypass on blood pressure variation. Please explain what is meant by “unsuitable monitoring” in line 49 on page 2.

Materials and Methods: Generally, there is a clear explanation of the study, data collection and analysis, although the reviewer has some concerns:

Please indicate if the analysis included a % change from baseline SBP and DBP. The term “types of surgery” is very broad, it is based on surgical subspecialty rather than type of procedure. Please be more specific and indicate what kind of surgery was performed (major groups, i.e. trauma brain, spine surgery, etc.). Neurosurgery – were any of the procedures performed in permissive or planned hypotension or were these cases excluded? Ethical concerns: It is true that in majority of cases no approval of the Ethics Committee for an observational study is necessary, but the application should have been filed and a waiver should have been issued. Please provide the number of the waiver from the Bioethical Committee.

Results: This section is adequately described, but additional improvement is necessary:

Due to a large number of patients it is advisable to perform a subgroup analysis for each type of surgery. As a reviewer I would like to point out that majority of the monitoring is based on mean arterial pressure (MAP) and the guidelines are also based on MAP. Therefore, it is advisable to add MAP_Cv to the analysis as an additional column in Table 2 and 3. The MAP is not only used in studies cited and discussed by the authors (i.e. 11), but also is the mainstay of perioperative monitoring according to the following references:

Walsh M1, Devereaux PJ, Garg AX, Kurz A, Turan A, Rodseth RN, Cywinski J, Thabane L, Sessler DI. Relationship between intraoperative mean arterial pressure and clinical outcomes after noncardiac surgery: toward an empirical definition of hypotension.Anesthesiology. 2013 Sep;119(3):507-15. doi: 10.1097/ALN.0b013e3182a10e26.

Sessler DI1, Bloomstone JA2, Aronson S3, Berry C4, Gan TJ5, Kellum JA6, Plumb J7, Mythen MG8, Grocott MPW9, Edwards MR7, Miller TE10; Perioperative Quality Initiative-3 workgroup; POQI chairs, Miller TE, Mythen MG, Grocott MP, Edwards MR; Physiology group; Preoperative blood pressure group; Intraoperative blood pressure group; Postoperative blood pressure group. Perioperative Quality Initiative consensus statement on intraoperative blood pressure, risk and outcomes for elective surgery.Br J Anaesth. 2019 May;122(5):563-574. doi: 10.1016/j.bja.2019.01.013. Epub 2019 Feb 27.

Sessler DI1, Khanna AK2,3. Perioperative myocardial injury and the contribution of hypotension.
Intensive Care Med. 2018 Jun;44(6):811-822. doi: 10.1007/s00134-018-5224-7. Epub 2018 Jun 4.

Tables - abbreviations, such as Cv, ASA or IQR, ESC, ESA should be explained in a legend under the tables. Please indicate what is meant by regional or combined anesthesia in Table 1. The cases with “regional anesthesia only” should be analyzed separately. The authors should explain what does the ‘p’ value refer to in Table 2. Is it an intra-group comparison? Especially interesting is the type of surgery data as the three types are completely different in terms of requirements (i.e. hypotensive anesthesia in neurosurgery).

Discussion: The authors discuss their results adequately. They emphasize and further discuss the importance of a relationship between results obtained in the study and the outcome of patients with high blood pressure variation, both systolic and diastolic. The discussion seems to be focused on MAP which is not included on the analysis. Please reconsider.

Limitations: If the authors decide not to include MAP_Cv calculation please discuss the reason for it and add this in the limitations section. Moreover, another limitation of this analysis is its single-center character focusing only on one specific hospital and population served.

Conclusion: This section is clear and reflects the results obtained by the authors.

References: There are many current references that justify the study.

This research paper is valuable to the readership; however it must be improved before considering it for publication. Therefore, I recommend a minor revision.

With best regards

Author Response

Thank you for such a detailed assessment of our paper and recommendation for its publication with minor revision.

We rephrased the sentence in the abstract. We included definitions of high BPV in the abstract. We underlined the lack of non-cardiac studies in the field. We made distinction for cardiac surgery studies. We explained ‘unsuitable monitoring’, as requested. No, we did not attempt to assess the % change from baseline values. There were multiple types of surgeries within those three specialities. Many of them were mixed in their nature (e.g. neurotrauma, complex abdominal and pelvic resections in oncologic subjects, gastrointestinal surgeries with vascular interventions). Therefore, we decided not to reveal such a detailed data. None of the procedures was performed using permissive hypotension. This is a non-interventional observational study and there was no need for Ethics Committee approval. However, we submitted a request to the Ethics Committee for official information about the waiver. We performed a sub-analysis of BPV in terms of the type of surgery but not within particular group of operations, as we explained above. We also focused on the position in which the patients were operated. However, both of them were of low importance in mortality prediction. Interestingly, mortality was dependent on the patient’s risk only. MAP-CV was calculated and included in all analyses. We explained the abbreviations in the tables. We explained the details, as requested. We added the limitation regarding single-centre analysis. We believe that after inclusion of MAP analysis, the discussion is more appropriate and interesting to the reader.

Kind regards, the Authors

Reviewer 3 Report

Thank you for an opportunity to review manuscript entitled:” Intraoperative blood pressure variability predicts postoperative mortality in non-cardiac surgery – a prospective observational cohort study.” by A. Wiórek and ŁJ. Krzych.

This is interesting prospective, cohort study investigating an association of blood pressure (BP) variability on perioperative mortality in patients undergoing non cardiac surgery. Authors analyzed both systolic and diastolic pressure variability in large cohort of patients and subsequently constructed prediction models.  High variability predicted mortality particularly diastolic BP variability. This is certainly novel and interesting approach towards analysis of important factor influencing perioperative outcome, however, I have 3 main points of critique:

Even though it is prospective, cohort study authors should have obtained approval or opinion of local Research Ethics Board. It should clearly state that approval is not needed according to local regulation. It is important in the context of fact that most of jurisdictions require REB approval for prospective cohort studies. Authors mentioned in introduction that blood loss is one of the major contributors to blood pressure changes/variation. Indeed from clinical point of view this is very common factor contributing to BP variability. Why results and model do not include information regarding perioperative blood loss, transfusion and anemia? Lack of these factors makes prediction model incomplete. I see it as major deficiency of methodology. Discussion needs to be revised and simplified. In current form it is somewhat defragmented and does not give clear message. I would suggest a following scheme of discussion The main findings of your study How do your results compare to other similar studies (if existing)? In your references I don’t see seminal papers by WS Bettie or L. Sun Implications of your results and clinical significance Limitations of your study  

Once again, I would imagine that improving your model by adding information describing blood loss, transfusion and perioperative anemia would change discussion significantly and strengthened your manuscript.

In summary, this is very interesting paper, however several parts of it require major revision and subsequent repeated peer-review.

Author Response

Reviewer #3.

Thank you for your comments on the paper.

This is a non-interventional observational study and there was no need for Ethics Committee approval. However, we submitted a request to the Ethics Committee for official information about the waiver. There is always a big concern regarding a number of potential determinants or confounders and the outcome. In the perioperative period there is a plethora of clinical and laboratory data which could be included into the final analysis. In our paper we decided to find a balance between the number of variables which are necessary to draw reliable conclusions and the number of variables which generate a mess in the understanding the paper by the reader. None of the patients in our analysis had massive bleeding so inclusion of perioperative blood loss of requirement of transfusions was of no importance. Only 24 subjects were transfused during the procedures. Using the cut-off value of 13 g/dl for both sexes, pre-op anaemia was diagnosed in 142 patients but there was no association between anaemia and the requirement for transfusion intraoperatively (p=0.4). BPV was higher in patients with anaemia, but this association was of no importance in strata of ASA-PS classes. Finally, the AUC of the final regression is very high (almost 0.9!), so we may assume that our statistical analysis covers major determinants of mortality. However, to come up to your expectations, we included this extremely interesting issue into study limitations. We modified the discussion according to your suggestions and included the papers of WS Beattie and L. Sun, as requested.      

Kind regards, the Authors

Reviewer 4 Report

To:

Editorial Board

Title: “Intraoperative blood pressure variability predicts postoperative mortality in non-cardiac surgery – a prospective observational cohort study”

Dear Editor,

I read this manuscript and I think that:

Types of drugs adopted for anaesthesia should be described as they can influence blood pressure. Such data should enter the multivariate regression model. The intraoperative management of fluids is a further element to be considered for the final analysis as it can influence blood pressure. The authors should discuss such a point. How did the authors exclude the influence of fluids management on final results? Comorbidities had not been taken into account for the evaluation of mortality. Why? How did the authors exclude the influence of them on final results? Table 1 should be updated with all of the clinical and instrumental characteristics of the study population. What about the inclusion and exclusion criteria? The authors should better discuss such a point. A flow chart of the study should be added.

Author Response

Reviewer #4.

Thank you for comments. They are critical but we feel a little confused because all of the requested data had been included in the paper before.

1. Types of drugs used for anaesthesia are included as ‘type of anaesthesia’. All patients received opioids. All TIVA patients received propofol and TCI was applied. Sevoflurane was the only volatile agent used in our subjects and MAC was used to adjust the dose. Only 7 patients received etomidate for induction. Regional anaesthesia included bupivacaine only. No other drugs were used in our cohort. There was no impact of anaesthesia on mortality (Table 4) 2. In our patients we used MAP-guided GDT to adjust the dose of fluids to maintain haemodynamic equilibrium. So, the fluid regimen was tailored strictly to the patients’ needs and requirements. However, to come up to your expectations, we included this extremely interesting issue into study limitations. 3. Comorbidities and the level of their clinical impact on physical status are assessed using ASA-PS system. 4. The inclusion and exclusion criteria are described in ‘Methods’. 5. The flow chart is depicted as figure 1.  

Kind regards, the Authors

Round 2

Reviewer 3 Report

Thank you for an opportunity to review a revised version the manuscript.

It is much improved version, however some concerns remain. I am only partially satisfied with response of authors and most likely they need to address concerns in somewhat advanced approach. Will summarize my comments once again but please have a look at my original comments once again:

Please remove paragraphs, which belongs to discussion from your introduction Please, clearly state that your REB does not require to issue an approval. These document should've accompanied your submission Part of your discussion still requires some work. It is not enough to put ref. of Sun et al. You need to discuss their results and conclusions. The same comment applies to preoperative anemia. Please expand on it.

Author Response

Thank you kindly for acknowledging the improvement of the paper with the corrections we have already made. As suggested we followed through with your further suggestions. Below please find the answers to the points you’ve made.

1.We modified the introduction. We have removed parts of the introduction regarding other similar studies in the subject and transferred them to the discussion section.  2. We clearly explained the reader that according to the Act on Medical Profession in Poland our REB did not require to issue an approval. 3. We expanded on the discussion of the two studies performed by Vernooij and Sun. We explained their results and conclusions to the reader. 4. We expanded on the subject of preoperative anaemia, according to your request.

Kind regards, the Authors

Reviewer 4 Report

The authors well addressed my previous comments. The paper improved very much as well as its scientific background.

Round 3

Reviewer 3 Report

Thank you for an opportunity to review next version of the manuscript.

I don't have any further comments.

Text requires only minor editing of English